# Handling Imbalanced Pseudolabels for Vision-Language Models with Concept Alignment and Confusion-Aware Calibrated Margin

**Yuchen Wang** [1]  **Xuefeng Bai** [1]  **Xiucheng Li** [1]  **Weili Guan** [1]  **Liqiang Nie** [1]  **Xinyang Chen** [1]

## Abstract

Adapting vision-language models (VLMs) to downstream tasks with pseudolabels has gained increasing attention. A major obstacle is that the pseudolabels generated by VLMs tend to be imbalanced, leading to inferior performance. While existing methods have explored various strategies to address this, the underlying causes of imbalance remain insufficiently investigated. To fill this gap, we delve into imbalanced pseudolabels and identify two primary contributing factors: concept mismatch and concept confusion. To mitigate these two issues, we propose a novel framework incorporating concept alignment and confusion-aware calibrated margin mechanisms. The core of our approach lies in enhancing underperforming classes and promoting balanced predictions across categories, thus mitigating imbalance. Extensive experiments on six benchmark datasets with three learning paradigms demonstrate that the proposed method effectively enhances the accuracy and balance of pseudolabels, achieving a relative improvement of 6.29% over the SoTA method. Our code is avaliable at https://github.com/Noahwangyuchen/CAP.

## 1. Introduction

Large vision-language models (VLMs; Radford et al., 2021; Li et al., 2022; 2021; Alayrac et al., 2022; Zhang et al., 2024a; Gao et al., 2024) pre-trained on extensive image-text pairs achieve remarkable performance across a wide range foundamental vision tasks, such as image classification (Zhou et al., 2021), semantic segmentation (Xu et al., 2022), and object detection (Gu et al., 2022). Nonetheless,

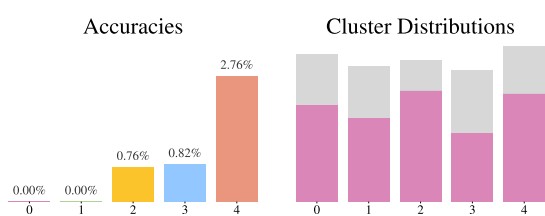

*Figure 1.* *left*: The lowest 5 per-class accuracies in RESISC45, *right*: The distribution of samples of them in clusters. The pink bar represents samples in the cluster in which they appear most frequently, the gray bar represents samples appear in other clusters.

previous research (Zhou et al., 2021; Gao et al., 2021; Zhang et al., 2022) shows that they still require adaptation using annotated data from downstream datasets to achieve optimal performance, which incurs substantial annotation costs.

Building upon the observation that VLMs, such as CLIP, inherently possess zero-shot image classification capabilities, previous studies (Huang et al., 2022; Menghini et al., 2023; Zhang et al., 2024b) explore adapting VLMs for downstream tasks by leveraging pseudolabels generated by the VLMs themselves. A critical challenge is that VLMs have biased preferences for different classes, which results in imbalanced pseudolabels, thus suffering from confirmation bias (Huang et al., 2022; Zhang et al., 2024b). While existing literature has explored strategies such as enforcing equal number of pseudolabels assigned to all classes (Huang et al., 2022; Menghini et al., 2023) and assigning a candidate set of pseudolables to each sample (Zhang et al., 2024b), limited research investigates the rationale behind the issue.

To fill this gap, we begin by studying the underlying causes of the imbalanced pseudolabels. We identify that the imbalance in pseudolabeling originates from the *semantic gap* (Xing et al., 2023) inherent in VLMs, where certain class names do not sufficiently correspond to visual concepts. To illustrate this, we identify the five classes with the lowest accuracies given by CLIP and visualize their per-class accuracies along with the cluster distribution of corresponding image features[1]. As shown in Figure 1, despite

---

[1]School of Computer Science and Technology, Harbin Institute of Technology (Shenzhen). Correspondence to: Xinyang Chen <chenxinyang95@gmail.com>, Xuefeng Bai <baixuefeng@hit.edu.cn>.

*Proceedings of the $42^{nd}$ International Conference on Machine Learning*, Vancouver, Canada. PMLR 267, 2025. Copyright 2025 by the author(s).

[1]We apply K-Means clustering to image features of samples in RESISC45 extracted by the image encoder of CLIP, forming 45 clusters since there are 45 classes in RESISC45.

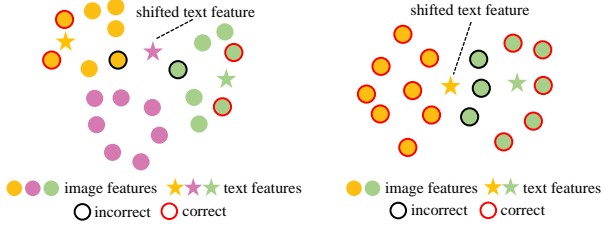

*Figure 2. left*: concept mismatch. *right*: concept confusion. Please see Appendix A for realistic examples of them.

the extremely low prediction accuracies for these classes, their image features exhibit good clustering performance, as they mostly concentrated in a single cluster. This indicates that CLIP fails to relate the name of certain classes to the corresponding visual concepts, thus resulting in imbalanced classification accuracies in zero-shot predictions.

To further study the issue of semantic gap, we delve into the erroneous classification results and identify that it leads to two key consequences, as depicted in Figure 2: *concept mismatch* and *concept confusion*[2]. Concept mismatch arises from a severe form of the semantic gap, where the text feature of a class is significantly misaligned with its corresponding image features. This leads to incorrect samples being assigned to the class during pseudolabeling based on prediction confidence, resulting in persistently low accuracy after fine-tuning. Concept confusion, on the other hand, is more commonly observed between similar classes, when the text features of the class names fail to capture the most distinguishable visual concepts between them. This leads to a bias seen in zero-shot prediction among these classes, causing an imbalance in prediction and pseudolabels.

To address these, we propose a concept-adaptive pseudolabeling (CAP) framework to fine-tune VLMs with balanced pseudolabels of unlabeled data on downstream tasks. To erase the concept mismatch, we propose a concept alignment strategy in which we employ iterative clustering to detect concept-mismatched classes and utilize large language models to generate enhanced textual descriptions. This approach aligns the text feature with the corresponding image features, ensuring that more correct samples are assigned to the class in the initialization stage. To tackle concept confusion, we introduce a confusion-aware calibrated margin that encourages the model to make more distinguishable predictions between similar classes and more balanced predictions over all classes, thereby improving the accuracy of pseudolabels. Moreover, we employ independent adapters on the visual branch to separately learn from highly reliable pseudolabels obtained through concept alignment and dynamically generated pseudolabels during training, thereby

---

[2]Take CLIP's zero-shot prediction for RESISC45 as an example, we discovered approximately 5% classes existing concept mismatch, and about 30% classes suffer from concept confusion, thus harming the accuracy.

mitigating the confirmation bias introduced by incorrect pseudolabels arising in the training process.

We conduct experiments on six image classification benchmarks across three learning paradigms, comparing our approach with previous methods. The results show that our framework consistently improves performance, achieving new state-of-the-art results. Further analysis confirms that the proposed method effectively mitigates the issues of concept mismatch and concept confusion, resulting in more balanced pseudolabels. The contributions of this work can be summarized as follows:

- We identify and analyze the causes of imbalance in VLMs' zero-shot predictions, attributing it to concept mismatch and concept confusion.
- We propose concept alignment and confusion-aware calibrated margin to address these issues, enhancing pseudolabel balance and accuracy.
- We conduct extensive experiments across unsupervised learning, semi-supervised learning, and transductive zero-shot learning, achieving a relative improvement of 6.29% over the SoTA method.

## 2. Related Works

**Vision-Language Models**. Recently, VLMs have demonstrated impressive performance on various downstream vision tasks, such as image classification (Zhou et al., 2021; Addepalli et al., 2024), semantic segmentation (Xu et al., 2022; Shi et al., 2024), and object detection (Gu et al., 2022; Kim et al., 2024; Wang et al., 2024). The key idea of VLMs is to learn representations that bridge the gap between visual and textual modalities, which facilitates general-purpose understanding and reasoning between modalities (Van den Oord et al., 2018), for example, CLIP (Radford et al., 2021), ALIGN (Li et al., 2021), Florence (Yuan et al., 2021). Despite great success, recent research (Zhou et al., 2021; Zhang et al., 2024a) indicates that a significant amount of labeled data remains crucial for adapting VLMs across various downstream tasks, which incurs substantial labeling costs. In this paper, we focus on fine-tuning CLIP – a widely adopted VLM, in downstream tasks with abundant unlabeled data, thus eliminating dependence on labeled data.

**Learning from Unlabeled Data**. In real-world downstream applications, obtaining a considerable amount of labeled data is expensive. With the zero-shot classification capability inherent in VLMs, recent work explores the use of pseudolabeled data for task adaptation. For instance, UPL (Huang et al., 2022), or FPL (Menghini et al., 2023) select top-$k$ confident samples for each class to form a balanced distribution among classes. Building upon this idea, GRIP (Menghini et al., 2023) exploits an iterative strategy, gradually increasing the value of $k$ with each iteration until all

unlabeled data are incorporated in the final iteration. Meanwhile, CPL (Zhang et al., 2024b) takes a similar iterative strategy to GRIP, assigning each sample a set of candidate pseudolabels each iteration, expecting the true label to be among them. Different from previous work which employ post-hoc methods to optimize pseudolabels, we identify the rationale behind the imbalanced pseudolabels and propose two approaches to address classes with inherently low accuracy, encouraging the model to make more balanced predictions among classes.

**Prompt Tuning.** This strategy was initially applied to large language models as a replacement for manually designed prompts and served as a fine-tuning method to provide task-specific information. With CoOp (Zhou et al., 2021) pioneering the application of prompt tuning for fine-tuning CLIP, this technique has been increasingly adopted in VLMs (Zhou et al., 2022; Bahng et al., 2022; Zang et al., 2022). Prompt tuning applied to different modalities has been explored. CoOp concatenate a group of continuous vectors to the input at the textual branch, optimizing these vectors to complete fine-tuning. VPT (Jia et al., 2022) introduces a small budget of additional parameters to the image encoder, which are prepended into the input sequence of each layer. MaPLe (Khattak et al., 2023) proposes a joint prompting approach by learning context prompts in the textual branch, and projecting them to the visual branch through a linear projection. In this paper, we adopt MaPLe as the prompt tuning method, as it allows learning from both modalities.

## 3. Methodology

**Problem Definition.** This paper studies how to adapt CLIP to downstream tasks without human-annotated data. Formally, given a collection of downstream unlabeled data denoted as $\mathcal{D}_{\mathrm{UL}} = \{(\boldsymbol{x}_i)\}_{i=1}^{N}$, comprising $N$ instances, we explore how to assign accurate pseudolabels to each instance within a classification label space $\mathcal{Y} = \{c\}_{c=1}^{C}$. The obtained datasets are then utilized to fine-tune CLIP, enhancing its applicability to the downstream tasks. We consider three learning paradigms: unsupervised learning (UL), semi-supervised learning (SSL) (Zhang et al., 2021; Chen et al., 2023; Wang et al., 2023), and transductive zero-shot learning (TRZSL), since all the paradigms explore the exploitation of unlabeled data.

**Motivation.** While VLMs have exhibited inherent zero-shot capabilities for pseudolabel generation, the *semantic gap* inherent in VLMs significantly restricts the accuracy of generated pseudolabel. We conduct an in-depth study on the *semantic gap* and observe that it manifests two phenomena: *concept mismatch* and *concept confusion*. In this paper, we first address *concept mismatch* through a detect-then-enhance framework; Subsequently, we alleviate *concept confusion* with a confusion-aware calibrated margin which

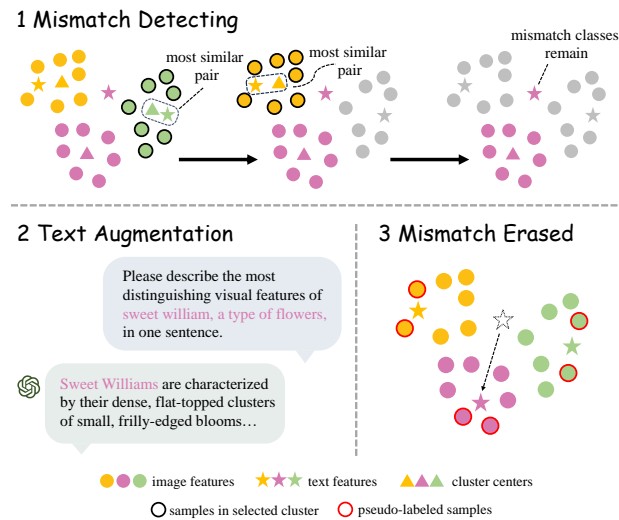

*Figure 3.* The process of concept alignment. We first take an iterative clustering strategy to detect the concept-mismatched classes. We then utilize LLMs to generate enhanced descriptions for them, and obtain images with top-$k$ similar image features to the enhanced text feature each class as pseudolabeled samples.

guides VLMs to generate more distinguishable predictions.

**Overview.** The overall workflow of our method can be divided into three steps: 1) we employ concept alignment (§3.1) to refine pseudolabels for concept-mismatched classes by detecting them and enhancing their text descriptions via a large language model, aligning the descriptions with visual concepts; 2) we then alleviate concept confusion with a confusion-aware calibrated margin (§3.2). This margin, derived from similarity between classes and model prediction tendencies, encourages the model to produce more discriminative and balanced predictions; 3) we propose a fine-tuning framework (§3.3) that utilizes both the pseudolabeled data generated in the concept alignment step and the remaining unlabeled data. We deploy main and pseudo adapters on the visual branch to learn from pseudolabeled and unlabeled data separately, and use the confusion-aware calibrated margin to compute the loss.

### 3.1. Concept Alignment

As illustrated in Figure 1 and Figure 2, concept mismatch leads to markedly low accuracy for certain classes. Consequently, very few correct pseudolabels are assigned to these classes and the accuracy for these classes remains exceptionally low after fine-tuning. To address this issue, we propose a concept alignment process designed to assign more precise pseudolabels to the concept-mismatched classes. As depicted in Figure 3, the process initiates with a mismatch detection algorithm that iteratively excludes well-matched classes, thereby isolating the concept-mismatched instances and their corresponding labels. Subsequently, it utilize a

**Algorithm 1** Mismatch Detection

**Input:** Image feature set $\mathcal{I} = \{\boldsymbol{v}_i\}_{i=1}^N$, text feature set $\mathcal{T} = \{\boldsymbol{w}_j\}_{j=1}^M$, class labels $\mathcal{Y} = \{c\}_{c=1}^C$, threshold $t$

**Output:** Remaining image feature set $\mathcal{I}_{\text{final}}$, remaining class labels $\mathcal{Y}_{\text{final}}$

**while** $|\mathcal{Y}| \geq t$ **do**

$\quad \mathcal{C} = \{\boldsymbol{c}_j\}_{j=1}^{|\mathcal{T}|} = \text{KMeans}(\mathcal{I}, |\mathcal{T}|)$

$\quad \mathbf{S}_{ij}^{\mathcal{TC}} = \text{sim}(\boldsymbol{w}_i, \boldsymbol{c}_j), \quad \forall \boldsymbol{w}_i \in \mathcal{T}, \boldsymbol{c}_j \in \mathcal{C}$

$\quad \mathbf{P}_{i,:}^{\mathcal{TC}} = \text{softmax}(\mathbf{S}_{i,:}^{\mathcal{TC}})$

$\quad (i^*, j^*) = \underset{i,j}{\text{argmax}}\ \mathbf{P}_{ij}^{\mathcal{TC}}$

$\quad \mathcal{I}_{j^*} = \{\boldsymbol{v}_k \mid \boldsymbol{v}_k \in \mathcal{I},\ k \in \text{Cluster}_{j^*}\}$

$\quad \mathcal{T} \leftarrow \mathcal{T} \setminus \{\boldsymbol{w}_{i^*}\},\ \mathcal{I} \leftarrow \mathcal{I} \setminus \mathcal{I}_{j^*},\ \mathcal{Y} \leftarrow \mathcal{Y} \setminus \{i^*\}$

**end while**

$\mathcal{I}_{\text{final}} = \mathcal{I}, \quad \mathcal{Y}_{\text{final}} = \mathcal{Y}$

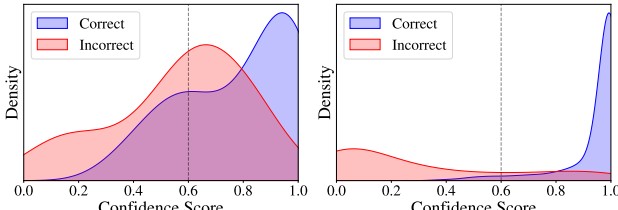

*Figure 4.* Density curve of confidence score for samples in concept-confused groups by *left*: zero-shot CLIP and *right*: CLIP fine-tuned with confusion-aware calibrated margin.

large language model to enhance text descriptions, ensuring more accurate alignment with their respective visual concepts. In this way, it reduces the occurrence of mismatches between visual data and their associated pseudolabels.

Given the text encoder $\psi$ and the image encoder $\phi$ of CLIP, we obtain the image feature $\boldsymbol{v}_i$ of each image in $\mathcal{D}_{\text{UL}}$ and text feature $\boldsymbol{w}_c$ of each class label with the template "a photo of a [CLS]", forming:

$$\mathcal{I} = \{(\boldsymbol{v}_i)\}_{i=1}^N, \quad \boldsymbol{v}_i = \phi(\boldsymbol{x}_i),$$

$$\mathcal{T} = \{(\boldsymbol{w}_c)\}_{c=1}^C, \quad \boldsymbol{w}_c = \psi(\text{template}(c)).$$

We started by detecting the concept-mismatched classes by an iterative clustering algorithm based on $\mathcal{I}, \mathcal{T}$, and $\mathcal{Y}$. The mismatch detection algorithm is presented in Algorithm 1. In this algorithm, we gradually remove the image features and text features of well-matched classes, thereby retaining only the concept-mismatched classes. For each iteration, we begin by applying K-means clustering to $\mathcal{I}$, forming $|\mathcal{T}|$ clusters and obtain the centroids $\mathcal{C}$. We then compute the similarity matrix $\mathbf{S}^{\mathcal{TC}}$ and probability matrix $\mathbf{P}^{\mathcal{TC}}$ of $\mathcal{T}$ and $\mathcal{C}$. Finally, we figure out the pair of text feature and centroid with the highest confidence score, $\boldsymbol{w}_{i^*}$ and $\boldsymbol{c}_{j^*}$, and we remove the image features in the cluster corresponding to $\boldsymbol{c}_{j^*}$ from $\mathcal{I}$, the text feature $\boldsymbol{w}_{i^*}$ from $\mathcal{T}$, the corresponding class label $i^*$ from $\mathcal{Y}$, since we assume that $i^*$ relates to the best-matched class in this iteration.

The algorithm terminates when the size of $\mathcal{Y}$ falls below a predefined threshold $t$. We denote the remaining image features and class labels as $\mathcal{I}_{\text{final}}$ and $\mathcal{Y}_{\text{final}}$. We further obtain the classes with the fewest-$t$ samples predicted to be as $\mathcal{Y}_{\text{low}-t}$, and finally identify the concept-mismatched classes as $\mathcal{Y}_{\text{MM}} = \mathcal{Y}_{\text{final}} \cap \mathcal{Y}_{\text{low}-t}$.

For the classes in $\mathcal{Y}_{\text{MM}}$, we then perform text augmentation

using a large language model (LLM) to generate enhanced text descriptions. Specifically, for each class $c$ in $\mathcal{Y}_{\text{MM}}$, we query the LLM $n$ times to generate $n$ corresponding descriptions. Akin to a step in Alg. 1, we identify the optimal description as the one that exhibits the highest similarity to one of the centroids derived from clustering on $\mathcal{I}_{\text{final}}$.

Finally, for each class in $\mathcal{Y}_{\text{MM}}$, we assign corresponding images of the image features with top-$k$ cosine similarity to the text feature of enhanced description as pseudolabeled samples of this class. For each class in $\mathcal{Y} \setminus \mathcal{Y}_{\text{MM}}$, we follow previous work (Menghini et al., 2023) by assigning pesudolabels based on the top-$k$ confidence scores obtained through zero-shot CLIP. We denote pseudolabeled samples generated this stage as $\mathcal{D}_{\text{PL}} = \{(\boldsymbol{x}, \tilde{y})\}$ with size $M = k \times C$. Please refer to Appendix B for more details of concept alignment.

### 3.2. Confusion-Aware Calibrated Margin

Confusion frequently arises among similar classes, impeding the model's ability to distinguish these classes. This further leads to biased predictions that lean towards one class, disrupting the balance of the pseudolabels. To address this, we propose a confusion-aware calibrated margin inspired by logit adjustment (Menon et al., 2021), which gradually reduces concept confusion by improving local calibration among confused groups.

As shown in Figure 4, zero-shot CLIP often makes incorrect predictions with high confidence due to concept confusion. A 0.6 threshold for pseudo-labeling would introduce many errors and amplify confirmation bias. In contrast, our confusion-aware calibrated margin provides local calibration, reducing confidence for incorrect predictions, making pseudo-labels selected at the same threshold more accurate and improving learning stability.

Specifically, we compute the confusion-aware calibrated margin based on the similarity of classes and the model's prediction tendency. Given an instance $\boldsymbol{x}$, we obtain the logit output by CLIP as $\boldsymbol{z}$. With the instance's label $y$, the confusion-aware calibrated margin can be defined as a variant of cross-entropy loss:

$$\mathcal{L}_m(y, \boldsymbol{z}) = -\log \frac{e^{z_y}}{e^{z_y} + \sum_{c \neq y} e^{z_c + \mathbf{M}_{yc}}}, \quad (1)$$

where $\mathbf{M}$ is the margin matrix, which is constructed from the similarity matrix $\mathbf{S}$ and the class-wise margin scales $\boldsymbol{m}$.

We start by computing the similarity matrix $\mathbf{S}$ of all classes. Given CLIP with learnable parameters $\theta$, the image features of samples with pseudolabel $c$ are calculated as

$$\mathcal{I}_c = \{\boldsymbol{v}_j \mid \boldsymbol{v}_j = \phi_\theta(\boldsymbol{x}_j),\ (\boldsymbol{x}_j, c) \in \mathcal{D}_{\mathrm{PL}}\}.$$

We then compute the prototypes of all classes to further obtain their similarity. The visual prototypes $\mathcal{I}$ are computed as the average of all image features corresponding to a certain class, and the text prototypes $\mathcal{T}$ are text features extracted by the model:

$$\mathcal{I} = \{(\overline{\boldsymbol{v}}_c)\}_{c=1}^{C}, \quad \overline{\boldsymbol{v}}_c = \mathrm{avg}(\mathcal{I}_c).$$

$$\mathcal{T} = \{(\boldsymbol{w}_c)\}_{c=1}^{C}, \quad \boldsymbol{w}_c = \psi_\theta(c).$$

Then, we obtain the similarity matrix $\mathbf{S}$ by computing the maximum similarity between visual prototypes and textual prototypes for each pair of classes as

$$\mathbf{S}_{ij} = \max(\mathrm{sim}(\overline{\boldsymbol{v}}_i, \overline{\boldsymbol{v}}_j), \mathrm{sim}(\boldsymbol{w}_i, \boldsymbol{w}_j)). \tag{2}$$

To determine the class-wise margin scales $\boldsymbol{m}$, we first compute $\sigma(c)$, the number of samples in $\mathcal{D}_{\mathrm{PL}}$ that are classified as class $c$ with a confidence exceeding a threshold $\tau$:

$$\sigma(c) = \sum_{i=1}^{M} \mathbb{I}(\max(\boldsymbol{p}_i) \geq \tau) \cdot \mathbb{I}(\arg\max(\boldsymbol{p}_i) = c). \tag{3}$$

Based on $\sigma(c)$, the model's class-wise tendency $\delta_c$ and the overall imbalanced degree $\Delta$ can be calculated as

$$\delta_c = 1 - \frac{\sigma(c)}{\max(\sigma(j))}, \quad c, j \in \mathcal{Y} \tag{4}$$

$$\Delta = \max_c(\delta_c) \tag{5}$$

For all classes, we define the class-wise margin scale as

$$\boldsymbol{m} = (m_1, m_2, \ldots, m_C)^\top, \quad m_c = m \times \Delta \times \delta_c \tag{6}$$

where $m$ is a predefined margin scale.

Finally, we compute margin matrix $\mathbf{M}$ using similarity matrix $\mathbf{S}$ obtained in Eq. 2 and class-wise margin scale $\boldsymbol{m}$ obtained in Eq. 6 by

$$\mathbf{M} = \mathbf{S} \odot \boldsymbol{m}, \tag{7}$$

where $\odot$ represents the hadamard product. The margin matrix $\mathbf{M}$ is a combination of inter-class similarity and the class-wise prediction tendency of the model, thus providing a margin that adaptively adjusts between classes based on their similarity and the model's prediction behavior. By incorporating $\mathbf{M}$ to cross-entropy loss, as defined in Eq. 1,

it encourages the model to make more confident predictions between classes with high similarity and in classes with low tendency, thus enhancing the model's discriminative ability for these classes. It should be noted that the margin matrix $\mathbf{M}$ is updated at each epoch, thereby facilitating the progressive alleviation of concpet confusion in the generated pseudolabels and improving their accuracy.

### 3.3. Fine-Tuning with Pseudolabels

Figure 5 illustrates the overall fine-tuning framework of our method. We use MaPLe (Khattak et al., 2023) as our prompt tuning strategy, which learns context prompts separately each layer from layer 1 to layer $L$, from both textual and visual branches of CLIP. Different from GRIP (Menghini et al., 2023) and CPL (Zhang et al., 2024b) which only train on pseudolabeled data generated in initialization, we also utilize the remaining unlabeled data by dynamically generating pseudolabels for them during training. Thus, we separate the training data into two parts, the unlabeled samples $\mathcal{D}_{\mathrm{UL}}$ and the pseudolabeled samples $\mathcal{D}_{\mathrm{PL}}$, as $\mathcal{D}_{\mathrm{PL}}$ is obtained in Section 3.1.

To better utilize the high label accuracy of $\mathcal{D}_{\mathrm{PL}}$ and the abundance of $\mathcal{D}_{\mathrm{UL}}$, we deploy the main adapter and the pseudo adapter on the visual branch. The main adapter learns only from $\mathcal{D}_{\mathrm{PL}}$ and is used to generate pseudolabels for $\mathcal{D}_{\mathrm{UL}}$, while the pseudo adapter learns from only $\mathcal{D}_{\mathrm{UL}}$ under the supervision of these pseudolabels. Since the main adapter only learns from $\mathcal{D}_{\mathrm{PL}}$ with highly accurate pseudolabels, this strategy avoids the accumulation of errors introduced by the pseudo adapter which learns from less accurate pseudolabels. We also deploy an adapter on the textual branch. It is worth noting that we disable all adapters in inference.

Formally, at each iteration, we have $\{(\boldsymbol{x}_i^{pl}, y_i^{pl})\}_{i=1}^{b}$ and $\{(\boldsymbol{x}_i^{u})\}_{i=1}^{b}$, where b is the batch size, as subsets of $\mathcal{D}_{\mathrm{PL}}$ and $\mathcal{D}_{\mathrm{UL}}$, respectively. We use $\psi^a$ to denote CLIP's text encoder with adapter, and $\phi^m$ and $\phi^p$ CLIP's image encoder with main adapter and pseudo adapter, respectively. For each sample in $\{(\boldsymbol{x}_i^{pl}, y_i^{pl})\}_{i=1}^{b}$, we compute the logit $\boldsymbol{z}^{pl}$ as

$$\boldsymbol{z}^{pl} = (z_1, z_2, \ldots, z_C),\ z_c = \mathrm{sim}(\phi^m(\boldsymbol{x}^{pl}), \psi^a(c)). \tag{8}$$

We then compute the loss on pseudolabeled samples using $\mathcal{L}_m$ defined in Eq. 1:

$$\mathcal{L}_{\mathrm{PL}} = \frac{1}{b} \sum_{i=1}^{b} \mathcal{L}_m(\boldsymbol{z}_i^{pl}, y_i^{pl}) \tag{9}$$

For samples in $\{(\boldsymbol{x}_i^{u})\}_{i=1}^{b}$, we first follow FixMatch (Sohn et al., 2020) to generate their corresponding pseudolabels, denoted as $\{(\hat{\boldsymbol{y}}_i^{u})\}_{i=1}^{b}$, using a confidence threshold $\tau$.

Next, we compute the logit as

$$\boldsymbol{z}^u = (z_1, z_2, \ldots, z_C),\ z_c = \mathrm{sim}(\phi^p(\Omega(\boldsymbol{x}^u)), \psi^a(c)), \tag{10}$$

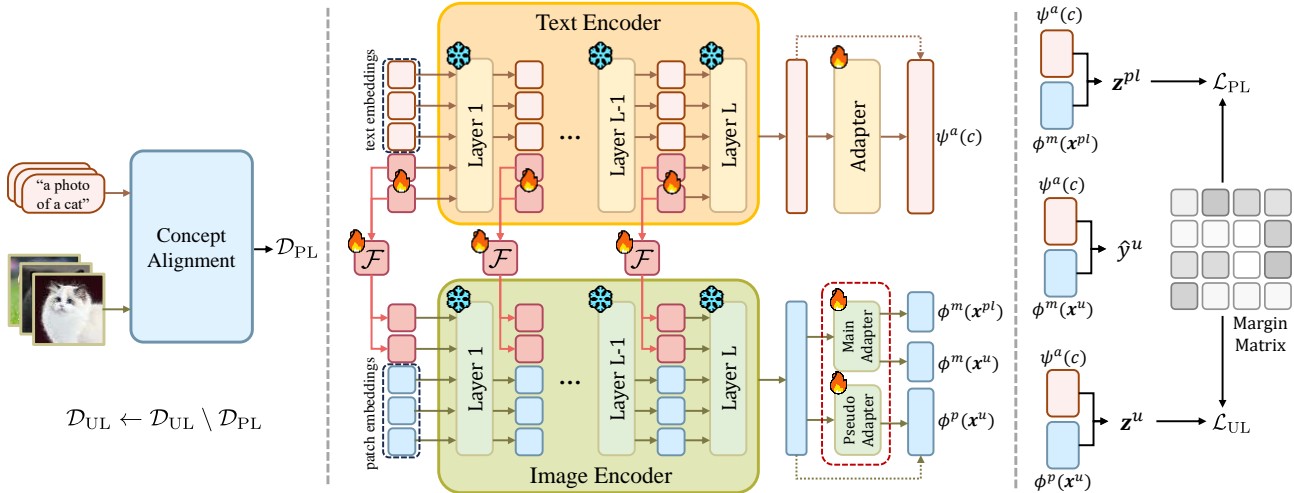

*Figure 5.* Overview of our framework. In the initialization stage, we use concept alignment to obtain $\mathcal{D}_{\text{PL}}$. In the fine-tuning stage, we deploy the main adapter and pseudo adapter to the visual branch, allowing for separate learning from pseudolabeled and unlabeled samples, and we utilize the confusion-aware calibrated margin matrix **M** to compute the loss.

where $\Omega$ represents image augmentation operation, and compute the loss on unlabeled data by

$$\mathcal{L}_{\text{UL}} = \frac{1}{b'} \sum_{i=1}^{b'} \mathcal{L}_m(\boldsymbol{z}_i^u, \hat{y}_i^u), \qquad (11)$$

where $b'$ is the number of samples with pseudolabels.

Finally, the overall loss is formulated as

$$\mathcal{L} = \mathcal{L}_{\text{PL}} + \mathcal{L}_{\text{UL}}. \qquad (12)$$

In the manner of semi-supervised learning and transductive zero-shot learning, we also have access to labeled samples $\{(\boldsymbol{x}_i^l, y_i^l)\}_{i=1}^b$. After we obtain $\boldsymbol{z}^l$ similarly in Eq. 8, we compute $\mathcal{L}_{\text{L}}$ as

$$\mathcal{L}_{\text{L}} = \frac{1}{b} \sum_{i=1}^b \mathcal{L}_m(\boldsymbol{z}_i^l, y_i^l), \qquad (13)$$

and add it to the overall loss:

$$\mathcal{L} = \mathcal{L}_{\text{PL}} + \mathcal{L}_{\text{UL}} + \mathcal{L}_{\text{L}}. \qquad (14)$$

## 4. Experiments

To examine the effectiveness of the proposed method, we conduct extensive experiments under three different learning paradigms across six benchmarks and compare our method with SoTA methods (§4.2). Moreover, we conduct ablation study (§4.3) and analysis (§4.4) to explore how the proposed method improve the performance on downstream tasks.

### 4.1. Experimental Settings

**Datasets.** We consider six image classification datasets covering diverse domains, including RESISC45 (Cheng et al., 2017), DTD (Cimpoi et al., 2014), EuroSAT (Helber et al., 2019), FGVC-Aircraft (Maji et al., 2013), CUB (Wah et al., 2011), Flowers102 (Nilsback & Zisserman, 2008).

**Learning Paradigms.** To ensure a thorough evaluation, we investigated three distinct learning paradigms: 1) unsupervised learning (UL) which provides with unlabeled data without any prior labels; 2) semi-supervised learning (SSL) which combines 2 labeled data per class with a larger pool of unlabeled data; 3) transductive zero-shot learning (TRZSL) in which classes are devided into seen classes with fully labeled data and unseen classes with fully unlabeled data. We set the ratio of seen to unseen classes at 62-38.

**Model Configuration.** In concept alignment, we set $t = \lceil \frac{C}{10} \rceil$ to determine the concept-mismatched classes. We use ChatGPT 4o-mini and set the query times $n = 5$ to obtain the enhanced descriptions. We set $k = 16$ to generate $\mathcal{D}_{\text{PL}}$. We set $m = 12$ as the predefined margin scale to compute the confusion-aware calibrated margin. Please refer to Appendix C.1 for more details.

**Baselines.** We compare our method with three existing methods, namely, Few Pseudolabels (FPL; Menghini et al., 2023), Grow and Refine Iteratively Pseudolabels (GRIP; Menghini et al., 2023), and Candidate Pseudo-Labeling (CPL; Zhang et al., 2024b). Since these methods can be applied with different prompting modalities, we report the results for the modality with the highest overall performance.

**Evaluation Metric.** Following CPL (Zhang et al., 2024b), we employ accuracy as the metric for evaluating model performance on test sets. We report the harmonic mean of the accuracies of seen and unseen classes in TRZSL. Specifically, we report the performance by calculating the test accuracy averaged over three seeds with standard deviation.

*Table 1.* Comparison results of test accuracy (%) on six benchmarks. The highest accuracies are bold.

| Methods | Flowers102 | | | RESISC45 | | | DTD | | |
|---|---|---|---|---|---|---|---|---|---|
| | SSL | UL | TRZSL | SSL | UL | TRZSL | SSL | UL | TRZSL |
| zero-shot CLIP | $63.67_{0.00}$ | | $63.40_{0.00}$ | $54.48_{0.00}$ | | $54.46_{0.00}$ | $43.24_{0.00}$ | | $43.45_{0.00}$ |
| FPL (Menghini et al., 2023) | $75.96_{0.74}$ | $65.67_{0.23}$ | $80.97_{0.00}$ | $68.13_{0.55}$ | $63.07_{0.38}$ | $72.11_{0.00}$ | $37.10_{5.45}$ | $44.96_{0.55}$ | $46.30_{0.03}$ |
| GRIP (Menghini et al., 2023) | $83.60_{0.48}$ | $69.84_{1.06}$ | $86.26_{0.00}$ | $74.11_{0.68}$ | $70.55_{0.88}$ | $81.07_{0.00}$ | $56.07_{0.85}$ | $46.09_{1.06}$ | $65.30_{0.01}$ |
| CPL (Zhang et al., 2024b) | $89.66_{0.36}$ | $72.90_{0.78}$ | $87.35_{0.76}$ | $80.98_{0.11}$ | $77.39_{0.44}$ | $85.85_{0.49}$ | $61.21_{0.56}$ | $51.91_{0.71}$ | $68.00_{0.34}$ |
| CAP (Ours) | $\mathbf{89.96}_{0.46}$ | $\mathbf{76.80}_{0.84}$ | $\mathbf{89.53}_{0.70}$ | $\mathbf{83.32}_{0.58}$ | $\mathbf{81.48}_{0.45}$ | $\mathbf{88.82}_{0.18}$ | $\mathbf{62.33}_{0.58}$ | $\mathbf{55.29}_{0.31}$ | $\mathbf{69.55}_{0.51}$ |

| Methods | EuroSAT | | | CUB | | | FGVCAircraft | | |
|---|---|---|---|---|---|---|---|---|---|
| | SSL | UL | TRZSL | SSL | UL | TRZSL | SSL | UL | TRZSL |
| zero-shot CLIP | $32.88_{0.00}$ | | $30.54_{0.00}$ | $51.82_{0.00}$ | | $51.57_{0.00}$ | $17.58_{0.00}$ | | $17.86_{0.00}$ |
| FPL (Menghini et al., 2023) | $62.05_{1.64}$ | $48.96_{1.49}$ | $53.70_{26.87}$ | $55.29_{0.59}$ | $53.04_{0.53}$ | $55.44_{0.20}$ | $20.02_{0.77}$ | $16.62_{0.67}$ | $17.55_{0.37}$ |
| GRIP (Menghini et al., 2023) | $58.66_{2.64}$ | $57.21_{1.77}$ | $92.33_{0.69}$ | $56.65_{0.33}$ | $51.42_{0.21}$ | $59.48_{0.38}$ | $16.98_{0.82}$ | $15.22_{0.71}$ | $26.08_{0.25}$ |
| CPL (Zhang et al., 2024b) | $77.51_{0.80}$ | $67.26_{0.47}$ | $93.78_{0.12}$ | $\mathbf{58.53}_{0.24}$ | $53.47_{0.36}$ | $\mathbf{66.20}_{0.50}$ | $\mathbf{22.48}_{0.63}$ | $18.35_{0.27}$ | $\mathbf{30.86}_{0.70}$ |
| CAP (Ours) | $\mathbf{92.78}_{0.34}$ | $\mathbf{75.01}_{1.94}$ | $\mathbf{96.64}_{0.27}$ | $58.04_{0.23}$ | $\mathbf{55.76}_{0.26}$ | $61.35_{0.05}$ | $21.79_{0.16}$ | $\mathbf{18.42}_{0.13}$ | $29.03_{0.49}$ |

## 4.2. Main Results

Table 1 compares the performance of different methods under vairous learning settings. We compare the proposed method with zero-shot CLIP, FPL, GRIP, and CPL across six datasets. It can be observed that the proposed method consistently surpasses existing methods under UL setting. Notably, our approach achieves significant improvements on Flowers102, RESISC45, and EuroSAT datasets, surpassing the CPL baseline by 3.90%, 4.09%, and 7.75%, respectively. These results underscore the efficacy of our method in leveraging unlabeled data to achieve superior classification outcomes. Moreover, our approach demonstrates competitive performance under the SSL setting across all datasets. This indicates that our method is capable of effectively integrating scarce labeled data with unlabeled data to enhance model performance. Under the TRZSL setting, our method surpasses CPL on Flowers102, RESISC45, DTD and EuroSAT by 2.18%, 2.97%, 1.55% and 2.86%, respectively. This suggests the capability of our method to leverage abundant labeled data with unlabeled data, further validating its versatility regarding different paradigms.

In addition, as illustrated in Figure 14, our method forms remarkably more balanced predictions compared to the baseline, which is inline with our motivation. Furthermore, our method is considerably less time-consuming than CPL and GRIP, achieving about 3.5 times the speedup over GRIP. Please refer to Appendix D.1 and D.2 for more details.

## 4.3. Ablation Study

**Ablation of Concept Alignment and Confusion-Aware Calibrated Margin.** To evaluate the effectiveness of each component of our method, we conducted an ablation study by independently removing each module and assessing the model's performance across three datasets under the UL setting. As shown in Table 2, both components individually result in noticeable performance improvements over the

*Table 2.* Ablation results of Concept Alignment (CA) and Confusion-Aware Calibrated Margin (CACM).

| CA | CACM | RESISC45 | DTD | Flowers102 |
|---|---|---|---|---|
| ✗ | ✗ | 68.41 | 49.57 | 70.85 |
| ✓ | ✗ | 72.48 | 53.13 | 72.58 |
| ✗ | ✓ | 78.77 | 52.76 | 74.49 |
| ✓ | ✓ | **82.03** | **55.26** | **76.77** |

*Table 3.* Ablation results of Independent Adapters (IA).

| Dataset | CPL | w/ IA | w/o IA |
|---|---|---|---|
| DTD | 51.9 | **55.3** | 54.6 |
| RESISC45 | 77.4 | **81.5** | 80.6 |
| EuroSAT | 72.9 | 76.2 | **78.3** |

baseline across all datasets. This underscores that each component independently facilitates better performance than the baseline. Compared with CA, CACM exhibits greater improvements on the RESISC45 and Flowers102 datasets, while achieving slightly lower results on the DTD dataset. A possible reason is that CLIP performs inferior extracting image features in DTD, leading to a substantial amount of mutual confusion among image features which is hard to address with prompt tuning. Moreover, the combination of both methods consistently yields the highest accuracy, highlighting their complementary effects in improving the accuracy of pseudolabels.

**Ablation of Independent Adapters.** In CAP, we deploy independent adapters on the visual branch to separately learn from $\mathcal{D}_{PL}$ and $\mathcal{D}_{UL}$, thus avoiding the accumulation of errors introduced by incorrect pseudolabels. To explore the effect of independent adapters, we report the test accuracy under UL setting across three dataset with and without independent adapters in Table 3. It can be observed that both model gives better results than CPL, and shared adapters gives generally comparable results to independent adapters.

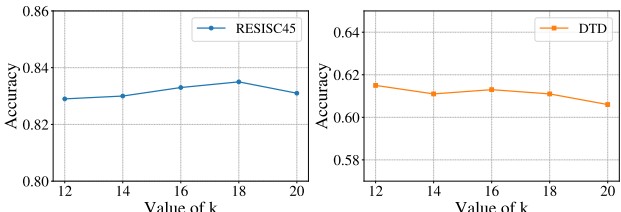

*Figure 6.* Ablation of different values of $k$ under SSL setting. We report the test accuracies on RESISC45 and DTD.

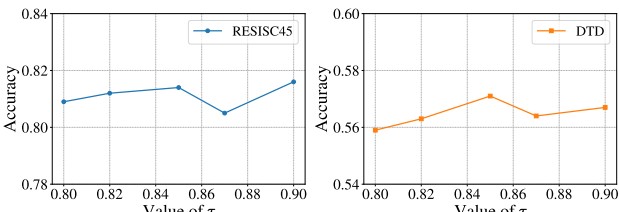

*Figure 7.* Ablation of different values of $\tau$ under UL setting. We report the test accuracies on RESISC45 and DTD.

**Ablation of Hyperparameters.** In CAP, two hyperparameters are directly related to the generation of pseudolabels. Specifically, $k$ determines the number of pseudolabels generated for each class in the initialization stage, while $\tau$ serves as the confidence threshold for dynamically assigning pseudolabels during training. We conduct ablation studies on different values of $k$ and $\tau$ using the RESISC45 and DTD datasets, and present the results in Figure 6 and Figure 7, respectively. The performance remains relatively stable across a wide range of values for both $k$ and $\tau$, indicating that CAP is generally robust to moderate fluctuations in these settings.

### 4.4. Analysis

**Does Concept Alignment Effectively Reduce Concept Mismatch?** The core of concept alignment lies in accurately detecting and correcting concept-mismatched classes, thereby improving the accuracy of pseudolabels. To validate this, we examined the accuracy of the pseudolabels corrected by our approach and compared it to the top-$k$ strategy employed by UPL (Huang et al., 2022) and GRIP (Menghini et al., 2023)[3]. The left sub-figure of Figure 8 visualizes the accuracy of generated pseudolabels for concept-mismatched classes in the Flowers102 dataset. As observed, the concept alignment approach results in consistently higher accuracies compared to the top-$k$ strategy, with a significant boost over 60% seen in four classes. This indicates that our proposed concept alignment mechanism effectively mitigates the concept mismatch issue. To further examine the impact of concept alignment, we evaluate the test accuracy after fine-tuning on downstream datasets under the UL setting. As shown in the right figure of Figure 8, concept alignment

---

[3]We also present the mismatch detection results, please refer to Figure 16 for detailed information.

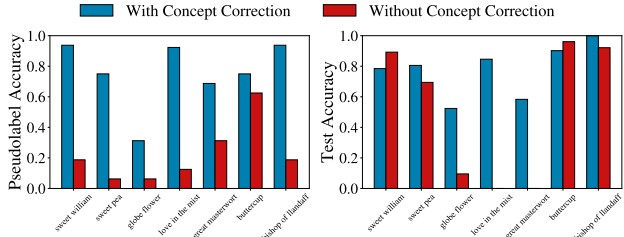

*Figure 8.* Evaluation of concept alignment. *left*: The accuracy of pseudolabels generated for concept-mismatched classes in Flowers102. *right*: The test accuracy after fine-tuning under UL setting.

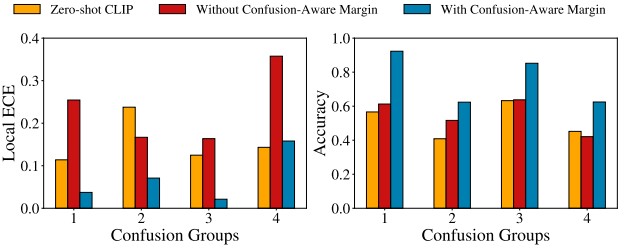

*Figure 9.* Evaluation of confusion-aware calibrated margin on RESISC45. *left*: Local ECE of concept-confused groups. *right*: Test accuracy of concept-confused groups after fine-tuning.

produces superior results compared to the top-$k$ method, notably within the classes "globe flower", "love in the mist", and "great masterwort". This confirms the effectiveness of concept alignment in improving pseudolabel quality and further enhancing overall performance.

**Does Confusion-Aware Calibrated Margin Effectively Reduce Concept Confusion?** We first evaluate its impact on local Expected Calibration Error (ECE) for different concept-confused groups in RESISC45. As illustrated in the left sub-figure of Figure 9, it is evident that directly fine-tuning the model using cross-entropy loss leads to a substantial increase in local ECE for most concept-confused groups. This indicates that naive fine-tuning approach results in poorer calibration and reduces reliability of the pseudolabels. In contrast, our proposed confusion-aware calibrated margin addresses this issue by explicitly promoting more distinguishable logits among concept-confused classes. By promoting greater separation among logits of similar classes, this mechanism helps the model avoid overconfident yet incorrect predictions. This results in notably lower local ECE values, reflecting improved calibration within the concept-confused groups, thus enhancing the accuracy and balance of pseudolabels. Furthermore, the right sub-figure of Figure 9 compares the downstream performance of different methods. Our approach consistently outperforms both zero-shot CLIP and fine-tuning without the calibrated margin across all concept-confused groups. This not only highlights the effectiveness of our method in improving calibration but also demonstrates its ability to enhance the model's discriminative ability.

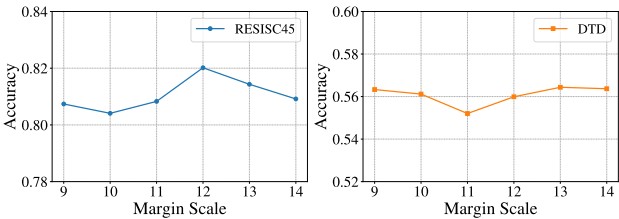

*Figure 10.* Evaluation of margin scale $m$. We report test accuracy of RESISC45 and DTD under UL setting.

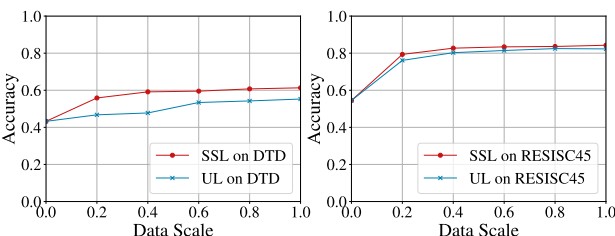

*Figure 11.* Performance on DTD and RESISC45 under SSL and UL settings with different proportion of unlabeled data.

*Table 4.* Results of test accuracy (%) using ViT-L/14 as the visual backbone. The highest accuracies are bold.

| | Methods | SSL | UL | TRZSL |
|---|---|---|---|---|
| **DTD** | Zero-shot CLIP | 52.45 | | 51.61 |
| | GRIP | 60.91 | 54.40 | 64.92 |
| | CPL | 69.82 | 57.20 | 71.97 |
| | CAP (Ours) | **70.26** | **65.85** | **73.61** |
| **RESISC45** | Zero-shot CLIP | 62.67 | | 62.13 |
| | GRIP | 81.53 | 76.86 | 86.88 |
| | CPL | 87.75 | 80.88 | 89.73 |
| | CAP (Ours) | **88.09** | **84.46** | **90.21** |
| **Flowers102** | Zero-shot CLIP | 73.98 | | 73.05 |
| | GRIP | 94.21 | 82.33 | 96.18 |
| | CPL | 96.80 | 83.94 | **97.34** |
| | CAP (Ours) | **98.09** | **84.19** | 95.40 |

**Analysis of Margin Scale.** In the confusion-aware calibrated margin, we use a predefined margin scale $m$ in computing the margin matrix, as shown in Eq. 6. Figure 10 presents the test accuracy under the UL setting for various values of $m$ on RESISC45 and DTD. Generally, the results show that the test accuracy remains relatively stable across different margin scales, suggesting that the proposed method is robust to variations in margin scale. Closer examination reveals a trend with drops at both ends. When $m$ is too small, the margin introduced to concept-confused groups becomes insufficient, making it difficult for the model to effectively resolve confusions between them. This results in a limited improvement in the distinguishability of the model between concepts-confused classes. Conversely, setting $m$ too large introduces excessively strong separation between class logits, which can lead to unstable training dynamics, harming the improvement of performance. Therefore, selecting a moderate value such as $m = 12$ offers a good trade-off between enhancing inter-class distinguishability and maintaining stable optimization dynamics.

**Impact of Data Scale.** To investigate whether our approach can efficiently leverage limited data, we evaluate our methods with different proportion of available unlabeled data. Figure 11 illustrates the impact of the scale of unlabeled data on the model performance under SSL and UL settings on DTD and RESISC45. The curves display a characteristic pattern of initial rapid growth followed by a gradual plateau, with around 0.6 and 0.4 on DTD and RESISC45, respectively. This indicates that while increasing the amount of unlabeled data leads to clear performance gains initially, the marginal benefit diminishes as the scale grows beyond a certain point. This pattern suggests that the model's performance experiences a rapid improvement as more unlabeled data becomes available, but eventually reaches a point where further increases in unlabeled data provide diminishing returns. The observed plateaus in performance indicate that the proposed method is highly efficient in utilizing unlabeled data, achieving substantial gains with relatively small amounts of unlabeled samples.

**Different Image Encoders.** To evaluate the generalizability of our method on different visual backbones, we conduct additional experiments on DTD, RESISC45 and Flowers102 utilizing a larger image encoder ViT-14/L as the visual backbone. As shown in Table 4, our method consistently outperforms other approaches across all three datasets. The performance demonstrates the robustness and adaptability of our method when integrated with different visual backbones, and that our method can effectively leverage the representational capacity of different vision models.

## 5. Conclusion

In this paper, we delved into the issue of imbalance in pseudolabels generated by VLMs, identifying two core underlying causes: concept mismatch and concept confusion. Building on the analysis, we proposed a novel framework incorporating concept alignment and confusion-aware calibrated margin to address two challenges. Our approach is capable of focusing on the underperforming classes and promoting balanced predictions across categories, thus improving the accuracy and balance of pseudolabels, leading to optimized performance. Extensive experiments on six benchmark datasets with three learning paradigms show that the proposed method effectively mitigates the issues of concept mismatch and concept confusion, resulting in more balanced and reliable pseudolabels, achieving a relative improvement of 6.29% over the SoTA method.

## Impact Statement

This paper presents work whose goal is to advance the field of Machine Learning. There are many potential societal consequences of our work, none which we feel must be specifically highlighted here.

## Acknowledgement

This work was supported by the National Natural Science Foundation of China (62306085, 62406091, 62206074, 62476071, 62236003), Shenzhen College Stability Support Plan (GXWD20231130151329002, GXWD20220811173233001), Guangdong Basic and Applied Basic Research Foundation (2025A1515012932).

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

# Appendix for
## *"Handling Imbalanced Pseudolabels for VLMs with Concept Alignment and Confusion-Aware Calibrated Margin"*

## A. Examples of Mismatch and Confusion

In this section, we provide realistic examples of *concept mismatch* and *concept confusion*. We show the t-SNE plots of image features extracted by CLIP from certain classes in RESISC45 dataset with their corresponding true labels and predicted labels generated by zero-shot CLIP.

Figure 12 shows examples of concept mismatch. It indicates that although the image features of these classes are relatively distinguishable, the samples represented as pink are completely misclassified. This illustrates that the text features of certain classes fail to capture the corresponding visual concepts, leading to significant semantic misalignment.

Figure 13 shows examples of concept confusion. The interwoven distribution of image features between these classes suggests a high degree of similarity. In the zero-shot prediction, most of samples are predicted to be a certain class, leaving relatively scarce samples predicted as the other. This indicates the text feature of the minority class fails to capture the most distinguishable visual concepts to align with corresponding image features, resulting an imbalanced prediction.

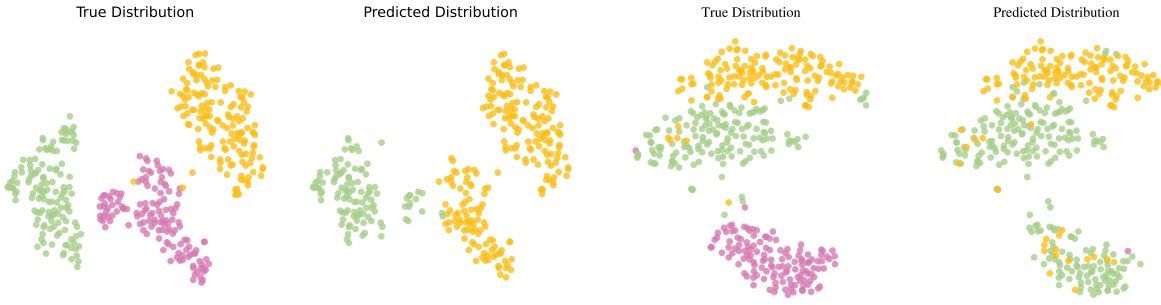

*Figure 12.* Two examples of *concept mismatch* in RESICS45. Pink represents the classes exisiting concept mismatch.

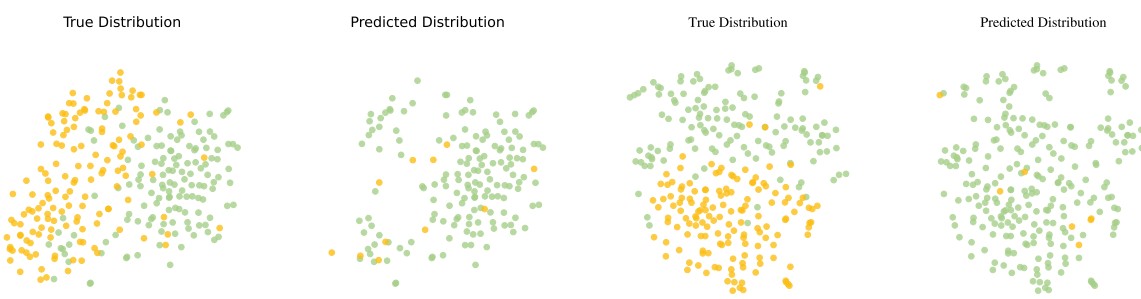

*Figure 13.* Two examples of *concept confusion* in RESICS45. Yellow represents the minority class.

## B. Details of Concept Alignment

In this section, we introduce more details of Alg. 1, and present examples of the prompt we use and the enhanced descriptions generated from LLMs.

### B.1. Details of Removing Samples

In Alg. 1, we remove the image features from the cluster corresponding to the class with best cross-modal alignment. However, the quantity of samples in this cluster might be too large, potentially containing samples for other classes. Thus,

we only remove top-$s$ samples predicted to be the best-matched class sorted by confidence, where $s$ is the average size of image features per class

$$s = \frac{|\mathcal{I}|}{|\mathcal{T}|}.$$

In each iteration, after we obtain $i^*$, we obtain the image features predicted to be class $i^*$ as

$$\mathcal{I}_{i^*} = \{\boldsymbol{v}_k \mid \boldsymbol{v}_k \in \mathcal{I}, \ \hat{y}_k = i^*\},$$

where $\hat{y}_k$ represents the predicted label of the sample corresponding to $\boldsymbol{v}_k$. Then, we obtain the intersection of $\mathcal{I}_{i^*}$ and $\mathcal{I}_{i^*}$, sort it by confidence score, and finally, select the top-$s$ samples as the samples to remove:

$$\mathcal{I}_{j^*} \leftarrow \mathrm{sort}(\mathcal{I}_{i^*} \cap \mathcal{I}_{j^*})[: s].$$

### B.2. Details of Text Augmentation

For text augmentation, we use the prompt template "Please describe the most distinguishing visual features of [CLS], in one sentence.". We present the enhanced descriptions of concept-mismatched classes detected in RESISC45 below as an example.

*Q: "Please describe the most distinguishing visual features of a satellite photo of chaparral, in one sentence."*

*A: "A satellite photo of chaparral features a dense, patchy, and irregular mosaic of shrubland vegetation, with a dominant brown or beige tone, and a speckled or mottled texture created by the mixture of shrubs, bare soil, and rocky outcrops."*

*Q: "Please describe the most distinguishing visual features of a satellite photo of terrace, in one sentence."*

*A: "A satellite photo of terrace is distinguished by a series of flat or gently sloping, rectangular or stepped areas, usually with distinct boundaries and varying tones or textures that distinguish the terraced fields."*

### B.3. Details of Identifying the Optimal Description

Given remaining image features $\mathcal{I}_{\mathrm{final}}$ and the corresponding text features of $n$ candidate descriptions of class $c$ denoted as $\mathcal{T}_c$, we first execute K-Means clustering to $\mathcal{I}_{\mathrm{final}}$ forming centroids $\mathcal{C}_{\mathrm{final}}$ by

$$\mathcal{C}_{\mathrm{final}} = \{\boldsymbol{c}_j\}_{j=1}^{|\mathcal{T}_c|} = \mathrm{KMeans}(\mathcal{I}_{\mathrm{final}}, |\mathcal{T}_c|).$$

We then compute the similarity matrix and probability matrix of $\mathcal{T}_c$ and $\mathcal{C}_{\mathrm{final}}$ and identify the pair with highest confidence score similar to Alg. 1, as

$$\mathbf{S}_{ij}^{\mathcal{TC}} = \mathrm{sim}(\boldsymbol{w}_i, \boldsymbol{c}_j), \quad \forall \boldsymbol{w}_i \in \mathcal{T}_c, \boldsymbol{c}_j \in \mathcal{C}_{\mathrm{final}}$$

$$\mathbf{P}_{i,:}^{\mathcal{TC}} = \mathrm{softmax}(\mathbf{S}_{i,:}^{\mathcal{TC}}),$$

$$(i^*, j^*) = \underset{i,j}{\mathrm{argmax}} \ \mathbf{P}_{ij}^{\mathcal{TC}}.$$

Finally, we select the description corresponding to $\boldsymbol{w}_{i^*}$ as the optimal enhanced description for class $c$.

### B.4. Details of the usage of $\mathcal{D}_{\mathrm{PL}}$

We add unlabeled data with high confidence into $\mathcal{D}_{\mathrm{PL}}$ to enhance the abundance of $\mathcal{D}_{\mathrm{PL}}$ during training. Specifically, we increase the size of $\mathcal{D}_{\mathrm{PL}}$ by $\frac{|\mathcal{D}_{\mathrm{UL}}|}{t}$ every 5 epochs. Denote $e$ as epochs to complete in one training, we compute $t$ as $t = \frac{e}{5}$. Once every 5 epochs, for each class label $c$, we obtain the unlabeled samples with predicted label $c$ with top-$\frac{|\mathcal{D}_{\mathrm{UL}}|}{t \times C}$ confidence, and add them into $\mathcal{D}_{\mathrm{PL}}$ as pseudolabeled samples.

*Table 5.* Detailed settings for experiments.

| | Flowers102 | RESISC45 | DTD | CUB | EuroSAT | FGVCAircraft |
|---|---|---|---|---|---|---|
| **Statistic data** | | | | | | |
| Class number | 102 | 45 | 47 | 200 | 10 | 100 |
| Training set size | 2040 | 6300 | 3760 | 5594 | 27000 | 6667 |
| Testing set size | 6149 | 25200 | 1880 | 5794 | 5000 | 3333 |
| **Training Setting** | | | | | | |
| Prompt Layers $L$ | 8 | | | | | |
| Prompt per Layer | 2 | | | | | |
| Image Augmentation | random resized crop | | | | | |
| Confidence Threshold $\tau$ | 0.85 | | | | | 0.5 |
| $k$ in top-$k$ strategy | 6 | 16 | | | | |
| Network | ViT-B / 32 | | | | | |
| Batch size | 32 | | | | | |
| Epoch | 50 where the first epoch is set for warmup | | | | | |
| Optimizer | SGD | | | | | |
| Momentum | 0.9 | | | | | |
| Learning rate (LR) | 0.01 | | | | | |
| Weight decay | 0.1 | | | | | |
| LR scheduler | CosineAnnealingLR | | | | | |

# C. Experimental Details

## C.1. Training Settings

We present the detailed training setting in Table 5.

## C.2. Comparison Methods

We briefly introduce the baselines in this section.

**Few-pseudolabels (FPL)** (Menghini et al., 2023): FPL is the same as UPL (Huang et al., 2022), which generates offline pseudolabels by selecting the top-$k$ confident samples per class in zero-shot predictions of CLIP.

**Grow and Refine Iteratively Pseudolabels (GRIP)** (Menghini et al., 2023): GRIP is built upon FPL, taking a iterative training strategy. For each Iteration, GRIP select top-$k$ confident samples per class in predictions by CLIP with soft prompts trained in last iteration. All soft prompts are re-initialized then, and GRIP start a new training iteration. Notably, the value of $k$ progressively increases after each iteration, and all unlabeled data will be included in the last iteration.

**Candidate Pseudolabel Learning (CPL)** (Zhang et al., 2024b): CPL takes a similar iterative strategy of GRIP, with a different strategy to select pseudolabels each iteration. CPL draw inspiration from the concept of the multiple annotations in crowdsourcing, constructing a set of potential true labels for model learning. CPL utilize inter-instance and intra-instance label selection to assign a set of candidate pseudolabels to a sample, and employ loss fuction designed for partial-label learning to update the soft prompts.

# D. More Experimental Results

## D.1. Effect of our method CAP

To illustrate the overview of the effect of our method CAP, we present the results after fine-tuning with and without CAP in Figure 14.

## D.2. Training Time

We present the time consumed training on EuroSAT with CAP (out method), CPL and GRIP in Figure 15. Our method takes significantly less time to complete fine-tuning, since the other two methods take an iterative strategy which executes the training process several times while our method only trains to converge once.

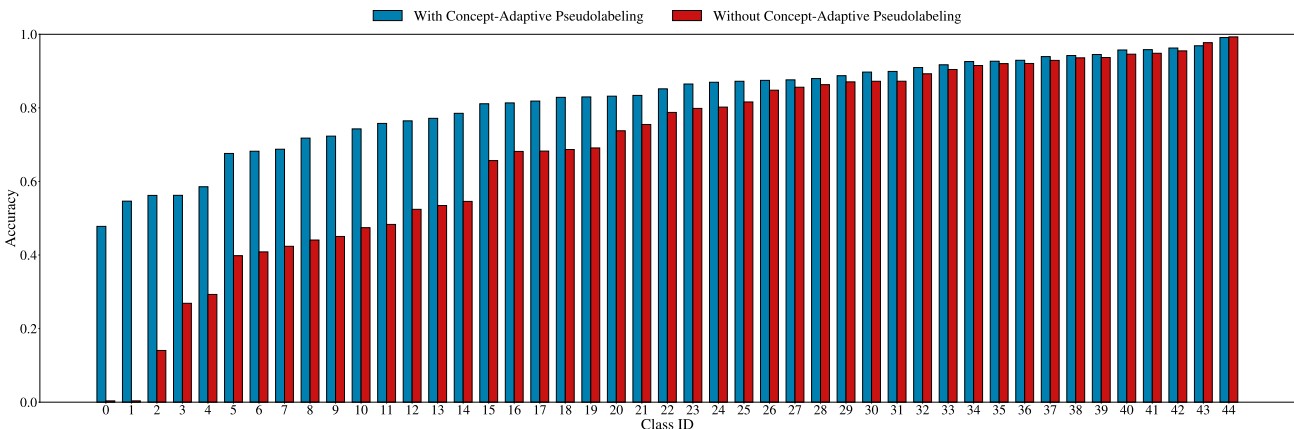

*Figure 14.* Test accuracy after fine-tuning with CAP and without CAP on RESISC45 under UL setting. Note that we still reserve the fine-tuning framework of CAP in the control group. It is clear that CAP forms a significantly more balanced prediction.

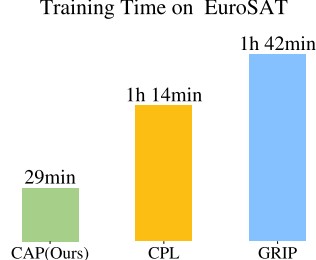

*Figure 15.* Time consumed training on EuroSAT with CAP, CPL and GRIP.

## D.3. Number of Mismatch Classes Detected

We present the number of concept-mismatched classes detected in Figure 16. Our approach achieves a significant performance improvement for previously underperforming classes while maintaining the exceptionally high accuracy of well-performing classes, demonstrating remarkable balance in model predictions.

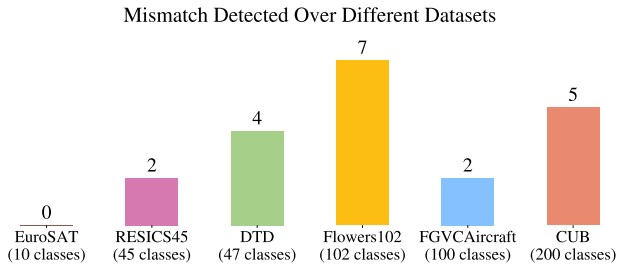

*Figure 16.* Visualization of the number of classes detected with concept mismatch over six datasets.

## D.4. More Evaluation of Confusion-Aware Calibrated Margin

The purpose of confusion-aware calibrated margin is to encourge CLIP to generate more distinguishable logits, thus gradually improve the balanced degree of CLIP's prediction among classes. In Figure 17, we illustrate the class-wise test accuracy on RESISC45 after fine-tuning under UL setting, highlighting the impact of incorporating the confusion-aware calibrated margin. From the left part of Figure 17, we observe that incorporating the confusion-aware calibrated margin significantly improves the accuracy of the lowest-performing classes. This indicates that the margin effectively mitigates the imbalance in predictions and reduces concept confusion among similar classes. On the right side of Figure 17, we see that the accuracy remains stable for well-predicted classes when applying confusion-aware calibrated margin. These results highlight the effectiveness of confusion-aware calibrated margin by addressing low-performing classes while maintaining high accuracy

for others.

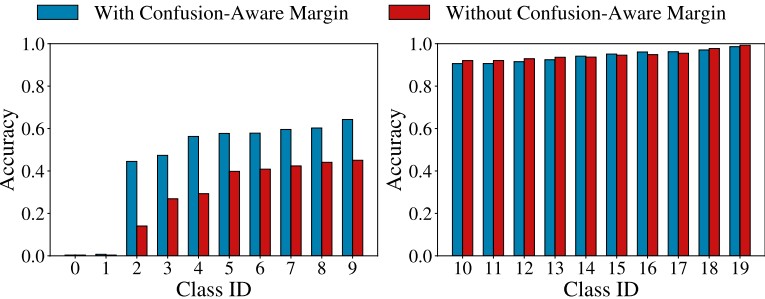

*Figure 17.* Class-wise test accuracy on RESISC45 under UL setting as an evaluation of confusion-aware calibrated margin. We disable concept alignment here. *left*: Visualization of lowest-10 class-wise accuracies. *right*: Visualization of highest-10 class-wise accuracies.

