# OpenReview forum: "Handling Imbalanced Pseudolabels for Vision-Language Models with Concept Alignment and Confusion-Aware Calibrated Margin"
_ICML.cc/2025/Conference — ICML 2025 poster_

### Official Review · Reviewer_xucA · 2025-03-06

**Overall Recommendation:** 4

**Summary:**

This paper proposes a novel approach to improving pseudolabels generated by vision-language models (VLMS). The authors identify two key errors which contribute to the degradation of pseudolabel quality: concept mismatch and concept confusion. Concept mismatch occurs when the class text name provides a description that is misaligned with the visual features. Concept confusion occurs when two (or more) classes contain significant overlap, and the text description fails to capture the most striking visual differences. To address these, the authors propose concept alignment, in which concept-mismatched classes are identified and their labels enhanced with an LLM, and the confusion-aware calibrated margin, in which a margin matrix is calculated per-class based on the inter-class similarity and class-wise detection confidence margin and integrated into the cross-entropy loss. The authors perform experiments in the unsupervised, semi-supervised, and transductive zero shot learning settings, and demonstrate improvement in fine-tuning across a wide variety of settings.

**Claims And Evidence:**

Yes, the authors claims are well supported empirically.

**Essential References Not Discussed:**

N/A

**Experimental Designs Or Analyses:**

Yes, the experimental design and analysis is valid.

**Methods And Evaluation Criteria:**

Yes, the benchmark datasets and various fine-tuning settings make sense for the task.

**Other Comments Or Suggestions:**

I find Figure 5 to be a little confusing; although the figure displays the entirety of the pseudolabel finetuning pipeline, the key components of CAP are not well highlighted. I think the figure should be simplified, and more emphasis should be placed on the components that are the authors contributions.

**Other Strengths And Weaknesses:**

Strengths:
- The paper is well-written and easy to follow
- CAP achieves good performance in a wide variety of experimental settings, consistently improving the SOTA.
- The method is well-motivated based on empirical observations, and detailed analysis (e.g. density of confidence scores in Fig 4) support the authors principle claims.

Weaknesses:
- CAP is largely heuristic and involves several differing hyperparameters, however detailed ablations of these are not included.

**Questions For Authors:**

N/A

**Relation To Broader Scientific Literature:**

Previous works, such as CPL, have investigated prompt-tuning to improve pseudolabeling by VLMs [A]. This work expands on this in a principled way through investigating what causes the misalignment between text and visual features, and directly addressing those with improvements.

[A] Candidate pseudolabel learning: Enhancing vision-language models by prompt tuning with unlabeled data. In Proc. of ICML.
OpenReview.net, 2024b

**Theoretical Claims:**

N/a

---

> ### Author Rebuttal · Authors · 2025-04-01
>
> Thanks for your time and encouraging comments! We address each of your concerns below.
>
> > **Q1:** Ablations of hyperparameters are not included
>
> Following your suggestion, we have conducted ablations of several hyperparameters in CAP.
>
> **Ablation of $t$**
>
> In CAP, we use $t$ to identify concept mismatched classes. Below, we report the results under UL setting with different values of $t$.
>
> *Table 1. Impact of threshold $t$.*
> |   | $t$ = $\frac{C}{14}$ | $t$ = $\frac{C}{12}$ | $t$ = $\frac{C}{10}$ (Default) | $t$ = $\frac{C}{8}$ | $t$ = $\frac{C}{7}$ |
> |----------|---------|---------|------------------|---------|---------|
> | **RESISC45** | 81.5 | 81.5 | 81.5 | 81.5 | 81.7 |
> | **DTD** | 54.9 | 54.9 | 55.4 | 55.4 | 55.4 |
>
> $C$ is the number of classes in the dataset. The results show that CAP maintains consistent performance across different threshold values.
>
> **Ablation of $k$**
>
> In CAP, we use $k$ to determine the number of pseudo labels generated for each class. Results of ablation of different values of $k$ under SSL setting are presented below.
>
> *Table 2. Impact of pseudolabel number $k$.*
> |   | $k$ = 12 | $k$ = 14 | $k$ = 16 (Default) | $k$ = 18 | $k$ = 20 |
> |----------|---------|---------|------------------|---------|---------|
> | **RESISC45** | 82.9 | 83.0 | 83.3 | 83.5 | 83.1 |
> | **DTD** | 61.5 | 61.1 | 61.3 | 61.1 | 60.6 |
>
> The results demonstrate that CAP is generally robust to different choices of $k$.
>
> **Ablation of $\tau$**
>
> In CAP, we use $\tau$ as the confidence threshold to dynamically generate pseudolabels. We conducted experiments on different values of $\tau$ under UL setting. The results are as follows:
>
> *Table 3. Impact of confidence threshold $\tau$.*
> |   | $\tau$ = 0.80 | $\tau$ = 0.82 | $\tau$ = 0.85 (Default) | $\tau$ = 0.87 | $\tau$ = 0.90 |
> |----------|---------|---------|------------------|---------|---------|
> | **RESISC45** | 80.9 | 81.2 | 81.4 | 80.5 | 81.6 |
> | **DTD** | 55.9 | 56.3 | 57.1 | 56.4 | 56.7 |
>
> The results confirm that our method is robust to changes in $\tau$.
>
> > **Q2:** Confusing Figure 5
>
> Following your valuable suggestion, we have revised Figure 5 to provide clearer visualization of CAP's key components while simplifying its overall presentation. The new figure is available at:
>
> https://anonymous.4open.science/r/CAP-C642/framework_updated.pdf
>
> We appreciate your insights and hope this revision improves clarity. We will incorporate all of the above experimental results and figures in the next version, and we welcome any additional feedback you may have!

---

### Official Review · Reviewer_YUBC · 2025-03-10

**Overall Recommendation:** 3

**Summary:**

This paper proposes a concept-adaptive pseudo-labeling framework to generate balanced pseudo-labels for fine-tuning vision-language models (VLMs) on downstream tasks.

In the first stage, the paper introduces concept alignment to address the issue of concept mismatch by assigning precise pseudo-labels to misclassified instances. In the fine-tuning stage, the paper proposes a confusion-aware calibrated margin, built upon logit adjustment, to mitigate concept confusion.

Experiments on six benchmarks demonstrate the effectiveness of the proposed approach, with ablation studies highlighting the contribution of each component.

## update after rebuttal

The authors have addressed my concerns, and I keep my initial positive rating.

**Claims And Evidence:**

Yes

**Essential References Not Discussed:**

No

**Experimental Designs Or Analyses:**

The proposed method is evaluated on six image classification benchmarks under SSL, UL, and TRZSL settings. Additional ablation studies on three datasets further demonstrate the effectiveness of the proposed components. Overall, the experimental setup and analysis are comprehensive and convincing.

One concern is the limited number (i.e., 4) of baselines for comparison. It would strengthen the paper if more baselines were included in the main results (i.e., Tab. 1).

**Methods And Evaluation Criteria:**

Overall, the technical contributions and evaluation are sound.

This paper focuses on fine-tuning VLMs for downstream tasks, specifically image classification, and conducts experiments on six benchmarks. The authors identify two key issues in generated pseudo-labels: concept mismatch and concept confusion. To address these issues, the paper proposes Concept Alignment (CA) and Confusion-Aware Calibrated Margin (CACM). The authors further support their approach with ablation studies under different datasets.

**Other Comments Or Suggestions:**

N/A

**Other Strengths And Weaknesses:**

Pros:
- Well-organized paper with clear motivation and strong writing.
- The proposed methods are effective and demonstrate SOTA performance.

Cons:
- As mentioned earlier, the number of compared baselines is limited—only four baselines are considered.
- The scope of evaluation is somewhat narrow. It would be more compelling if the method were tested on more challenging tasks, such as segmentation, rather than just image classification.

**Questions For Authors:**

See strengths and weaknesses.

**Relation To Broader Scientific Literature:**

This paper addresses the issues of mismatch and bias in VLM-generated pseudolabels, contributing to broader research areas, particularly in VLM-based training and pseudo-labeling techniques. The proposed approach has the potential to improve the reliability of pseudo-labeling for various downstream tasks, extending its impact beyond image classification.

**Theoretical Claims:**

N/A—this paper does not include theoretical proofs.

---

> ### Author Rebuttal · Authors · 2025-04-01
>
> Thanks for your time and helpful comments! We address each of your concerns below.
>
> > **Q1:** The number of baselines is limited
>
> In our paper, we primarily compare against state-of-the-art methods **specifically designed for leveraging CLIP’s zero-shot capabilities** through pseudolabel generation. The number of such methods is relatively small—for instance, CPL only compares with three baselines.
>
> To further enrich our comparisons, we have incorporated **general semi-supervised learning methods** as additional baselines, and the results under SSL setting are presented in the table below.
>
> *Table 1. Comparison of different methods under SSL setting.*
> |   | RESISC45 | DTD | Flowers102 | EuroSAT |
> |----------|---------|---------|------------------|---------|
> | **FixMatch [1]** | 66.1 | 51.2 | 85.0 | 79.1 |
> | **FreeMatch [2]** | 76.6 | 53.5 | 87.7 | 90.0 |
> | **SoftMatch [3]** | 69.5 | 47.8 | 86.9 | 83.8 |
> | **CAP (Ours)** | **83.3** | **62.3** | **90.0** | **92.8** |
>
> The results show that CAP consistently outperforms these baselines on all datasets, with a significant improvement of 6.7% on RESISC45 and 9.8% on DTD.
>
> [1] FixMatch: Simplifying Semi-Supervised Learning with Consistency and Confidence. In Proc. of NeurIPS, 2020.
> [2] FreeMatch: Self-adaptive Thresholding for Semi-supervised Learning. In Proc. of ICLR, 2023.
> [3] SoftMatch: Addressing the Quantity-Quality Trade-off in Semi-supervised Learning. In Proc. of ICLR, 2023.
>
>
> > **Q2:** The scope of evaluation is somewhat narrow
>
> We appreciate your suggestion to evaluate our method on more challenging tasks such as segmentation. However, our method is primarily designed to address the issue of pseudolabel imbalance caused by the image-level classification bias of VLMs, making it less directly applicable to segmentation tasks.
>
> A key requirement of our approach is the ability to extract image features for each class and use them for clustering or similarity computation at different stages of the algorithm. In segmentation, however, a single image **typically contains multiple classes**, prevents the extraction of clean, class-specific visual prototypes. This intrinsic difference in problem formulation currently limits the applicability of our method.
>
> In fact, existing works in top machine learning conferences that address this problem, such as CPL [4], LaFTer [5], and FineSSL [6], have similarly concentrated on classification tasks. This is because classification serves as a fundamental benchmark for evaluating improvements in pseudolabeling and adaptation strategies for VLMs.
>
> While extending these techniques to segmentation or other structured prediction tasks is an interesting direction, it would require task-specific modifications beyond pseudo-labeling (e.g. in segmentation, one often needs to design a more advanced segmentation network). That being said, we fully acknowledge the importance of extending these techniques to structured prediction tasks like segmentation, and we will include this promising avenue in our future work discussion.
>
> [4] Candidate pseudolabel learning: Enhancing vision-language models by prompt tuning with unlabeled data. In Proc. of ICML, 2024.
> [5] LaFTer: Label-Free Tuning of Zero-shot Classifier using Language and Unlabeled Image Collections. In Proc. of NeurIPS, 2023.
> [6] Erasing the Bias: Fine-Tuning Foundation Models for Semi-Supervised Learning. In Proc. of ICML, 2024.
>
> ---
>
> We will incorporate all of the above experimental results and analyses in the next version.

---

### Official Review · Reviewer_1DgE · 2025-03-12

**Overall Recommendation:** 3

**Summary:**

This paper proposes a novel framework, CAP (Concept-Adaptive Pseudolabeling), to address the problem of imbalanced pseudolabels when fine-tuning Vision-Language Models (VLMs) like CLIP for downstream tasks using unlabeled data. The authors identify two key causes of imbalance: concept mismatch (where text features of a class are misaligned with image features) and concept confusion (where similar classes are hard to distinguish). CAP tackles these issues with two main components: 1) a concept alignment strategy that iteratively detects and corrects concept-mismatched classes using LLMs to generate enhanced textual descriptions, and 2) a confusion-aware calibrated margin that encourages the model to make more distinguishable predictions between similar classes. The framework also employs independent adapters on the visual branch to learn from both highly reliable (concept-aligned) and dynamically generated pseudolabels. Extensive experiments across six image classification benchmarks and three learning paradigms (unsupervised, semi-supervised, and transductive zero-shot) demonstrate that CAP consistently improves performance, achieving a relative improvement of 6.29% over the state-of-the-art.

**Claims And Evidence:**

yes

**Essential References Not Discussed:**

NO

**Experimental Designs Or Analyses:**

The experimental designs are rational, and the results are convincing.

**Methods And Evaluation Criteria:**

yes

**Other Comments Or Suggestions:**

No

**Other Strengths And Weaknesses:**

Strength:

1. The identification of concept mismatch and concept confusion as distinct sources of imbalance is insightful and well-supported by the analysis and visualizations .

2. The confusion-aware calibrated margin provides a mechanism for improving local calibration and mitigating bias among confused classes.

3. The paper is well-written, clearly explains the proposed approach, and provides sufficient details for implementation

Weakness:

1. While the paper provides a good qualitative explanation of 'concept mismatch' and 'concept confusion,' a more formal or quantitative definition of these concepts would be beneficial.

2. While the paper demonstrates improved training time compared to CPL and GRIP, a more detailed breakdown of the computational cost of each component of CAP (e.g., concept alignment vs. fine-tuning) would be useful.

3. A more thorough discussion of the limitations would benefit the paper

**Questions For Authors:**

See strength and weakness part.

**Relation To Broader Scientific Literature:**

The paper builds upon a growing body of work on adapting VLMs to downstream tasks using pseudo-labels. It directly addresses the limitations of existing methods that suffer from imbalanced pseudo-labels due to confirmation bias.

**Theoretical Claims:**

yes

---

> ### Author Rebuttal · Authors · 2025-04-01
>
> Thanks for your time and helpful reviews! We address each of your concerns below.
>
> > **Q1:** A more formal or quantitative definition of concept mismatch and concept confusion
>
> Generally, concept mismatch arises from a severe form of the semantic gap while concept confusion is more commonly observed between similar classes. We provide more formal definitions below:
>
> **Concept Mismatch:** A class $y$ is considered to exhibit **concept mismatch** if the predicted label $\hat{y}$ for its visual prototype $\boldsymbol{v}_y$ (the average of visual features for class $y$) is **incorrect**, i.e.,
>
> $$
> \hat{y} = \arg\max p(\boldsymbol{v}_y) \neq y
> $$
>
> where $p(\boldsymbol{v}_y)$ is the predicted probability distribution over all classes for prototype $\boldsymbol{v}_y$.
>
> **Concept Confusion:** Given the confusion matrix **$\mathbf{M}$** predicted by CLIP, two classes $i$ and $j$ are considered to exhibit **concept confusion** if the misclassification between them is significant, specifically:
>
> $$
> \mathbf{M} _ {ij} + \mathbf{M} _ {ji} > 0.25 \times (N_i + N_j)
> $$
>
> where $N_i$ and $N_j$ are the total number of samples belonging to class $i$ and $j$, respectively.
>
> Based on these definitions, we report the number of classes exhibiting concept mismatch and concept confusion in RESISC45 and DTD.
>
> *Table 1: Number of classes exhibiting concept mismatch and concept confusion.*
> | Dataset | Concept Mismatch | Concept Confusion |
> | --------|------------------|-------------------|
> | **RESISC45** | 6 | 17 |
> | **DTD** | 9 | 24 |
>
> We will incorportate these definitions into the next version of our paper to improve clarity and precision.
>
> > **Q2:** Computational cost of CAP
>
> CAP, CPL, GRIP share two stages:
>
> 1. **Pseudolabeling stage** – Responsible for generating pseudo-labels, only needs to be run once per dataset.
> 2. **Fine-tuning stage** – Where the model is trained using the generated pseudo-labels.
>
> Additionally, CAP employ MismatchDetection algorithm to fix mismatched concepts.
>
> Below, we compare the computational cost of these two stages of training on EuroSAT for CPL, GRIP and CAP, using an NVIDIA RTX 4090 GPU and a Intel(R) Xeon(R) Silver 4314 @2.40GHz CPU:
>
> *Table 2: Computation time for each stage of CPL, GRIP and CAP.*
> | Method | MismatchDetection Time | Pseudolabeling Time | Fine-tuning Time |
> |--------|-------------------|------------------|------------------|
> | **CPL** | - | 3min 32s | 74min |
> | **GRIP** | - | 3min 32s | 102min |
> | **CAP** | 47s | 3min 32s | 29min |
>
> In can be observed that the mismatch detection algorithm of CAP takes about 47s. In fine-tuning, CPL requires more time as it takes an **iterative strategy requiring training to convergence multiple times** (10 iterations in the original implementation). In each iteration, CPL expand the pseudo-labeled dataset. GRIP shares a similar trend with CPL.
>
> *Table 3: Computation time for each iteration of CPL.*
> | Iteration of CPL | Training Time |
> |-----------|-------------------------|
> | **1st**  | 1 min |
> | **5th**  | 7 min |
> | **10th** | 13 min |
> | **Total** | 74 min |
>
> > **Q3:**  Discussion of the limitations
>
> One limitation of our method (CAP) is that it relies on a mismatch detection algorithm when handling **concept mismatch**. Currently, our detection method is a **simple clustering-based algorithm**, which, despite its computational efficiency, has room for improvement in detection accuracy and robustness.
>
> In future work, we plan to explore more **precise mismatch detection algorithms** to better identify  mismatched concepts, thereby further improving the robustness and accuracy of our method.
>
> ---
>
> We will incorporate all of the above experimental results and analyses in the next version.

---

> > ### Comment · Reviewer_1DgE · 2025-04-05
> >
> > Thanks , I keep my rate unchanged.

---

### Official Review · Reviewer_d39N · 2025-03-14

**Overall Recommendation:** 3

**Summary:**

The paper addresses concept mismatch and confusion when adapting VLMs to downstream tasks using pseudo-labeled data. The authors argue that this issue primarily arises from an imbalance in the pseudo-labels generated by VLMs. To mitigate this, they propose a concept alignment mechanism and a confusion-aware calibrated margin strategy. Additionally, they introduce independent adapters that separately process the original labeled data and pseudo-labeled data. Extensive and diverse experiments demonstrate the effectiveness of the proposed methods.

**Claims And Evidence:**

I believe the imbalance of pseudo-labels is a crucial challenge in adapting VLMs to downstream tasks using pseudo-labeled data. The proposed methods offer a reasonable solution to this issue, and their effectiveness is well-supported by appropriate experiments. The papers are well written to follow the core concept and motivation

- However, the ablation studies could be more comprehensive. Given the complexity of the method, additional components should be ablated, such as the choice of gamma, the use of independent adapters, and, if feasible, the selection of prompt tuning. Additionally, some abbreviations (e.g., CA, CACM) are not properly explained, which could enhance clarity.

- While the proposed method generally outperforms baselines, certain metrics, particularly on CUB TRZSL, show significant underperformance. Analyzing the root cause of this discrepancy would be needed.

- I wonder about the rationale behind (2). There can be various similarity calculation strategies.

**Essential References Not Discussed:**

To the best of my knowledge, there are no missing essential references, though I am not deeply familiar with this specific literature.

**Experimental Designs Or Analyses:**

The effectiveness of the proposed methods is well validated through various experiments (Figures 4, 6, 7, 8, and 9). Additionally, experiments conducted with different data scales and backbone sizes demonstrate the consistency of the methods across varying settings.

**Methods And Evaluation Criteria:**

Methods and evaluation criteria seem reasonable. As far as I know, the evaluation criteria follow the previous works (common benchmarks).

**Other Comments Or Suggestions:**

Described above

**Other Strengths And Weaknesses:**

Described above

**Questions For Authors:**

Described above

**Relation To Broader Scientific Literature:**

Considering the frequent use of pseudo labels in this literature, the proposed idea is worth to share

**Theoretical Claims:**

There are no theoretical claims.

---

> ### Author Rebuttal · Authors · 2025-04-01
>
> Thanks for your time and helpful comments! We address each of your concerns below.
>
> > **Q1:** The ablation studies could be more comprehensive
>
> Following your feedback, we have conducted additional ablation studies on the components you mentioned.
>
> **Ablation of gamma Selection**
>
> We conducted experiments on different values of $\tau$ (we assume that the reference to gamma was intended to refer to $\tau$, the confidence threshold) under UL setting. The results are as follows:
>
> *Table 1: Performance regarding $\tau$ under UL setting.*
> |   | $\tau$ = 0.80 | $\tau$ = 0.82 | $\tau$ = 0.85 (Default) | $\tau$ = 0.87 | $\tau$ = 0.90 |
> |----------|---------|---------|------------------|---------|---------|
> | DTD | 55.9 | 56.3 | 57.1 | 56.4 | 56.7 |
> | RESISC45 | 80.9 | 81.2 | 81.4 | 80.5 | 81.6 |
>
> It can be observed that our method is generally robust to changes in $\tau$.
>
> **Ablation of Independent Adapters**
>
> We evaluate our method with and without the independent adapters under UL setting. The results are as follows:
>
> *Table 2: Performance of different adapter configurations under UL setting.*
> |    | CPL (baseline) | w/ Independent Adapters | w/o Independent Adapters |
> |----------|---------|---------|---------|
> | DTD | 51.9 | 55.3 | 54.6 |
> | RESISC45 | 77.4 | 81.5 | 80.6 |
> | EuroSAT | 72.9 | 76.2 | 78.3 |
>
>
> It can be observed that both model gives better results than CPL, and shared adapters gives generally comparable results to independent adapters.
>
> **Impact of Prompt Tuning Methods**
>
> We replace MaPLe with two prompt tuning methods (i.e., UPT [1] and VPT [2]) and compare the performance under UL setting.
>
> *Table 3: Impact of different prompt tuning methods.*
> |   | DTD | EuroSAT |
> |----------|---------|---------|
> | **CAP w. UPT** | 54.9 | 77.5 |
> | **CAP w. VPT**  | 56.2 | 77.3 |
> | **CAP w. MaPLe** |  55.3 | 75.0 |
>
> The results demonstrate that CAP can be effectively integrated with various prompt tuning methods.
>
> [1] Unified Vision and Language Prompt Learning. ArXiv, abs/2210.07225, 2022.
> [2] Visual Prompt Tuning. In Proc. of ECCV, 2022.
>
> > **Q2:** Abbreviations (e.g., CA, CACM) are not properly explained
>
> Thank you for pointing this out. We apologize for the oversight. **CA** and **CACM** are abbreviations for **C**oncept **A**lignment and **C**onfusion-**A**ware **C**alibrated **M**argin. We will ensure that these abbreviations are properly explained in the next version.
>
> > **Q3:** Analysis of underperformance on CUB TRZSL
>
> Thanks for pointing this issue out. By analyzing the confusion matrices, we found that our method has more classes with zero accuracy than CPL. To further study this, we conduct a in-depth analysis of the results on CUB.
>
> CUB consists of 200 fine-grained bird species and exhibits **frequent concept mismatches**. For instance, CUB includes **four species of orioles**, but CLIP extracts **highly similar text features** for their category names. As a result, all four species are aligned with the visual features of a single dominant oriole species, leading to incorrect pseudo-labeling. Due to the **simplicity of our mismatch detection algorithm**, many of these mismatches go undetected, preventing affected classes from receiving correct pseudo-labels and limiting fine-tuning improvements.
>
> In contrast, CPL assigns multiple pseudo-labels to each sample. Since the misclassified categories are **semantically related**, there is **a higher chance that the correct label appears within CPL’s candidate label set**. This enables CPL to improve accuracy on these classes more effectively, which explains its superior performance under this particular setting. However, such extreme concept mismatches are relatively rare across datasets.
>
> > **Q4:** Rationale behind (2)
>
> **The choice of similarity:** In Equation (2), we use cosine similarity to measure the similarity between the text and visual prototypes of two classes, as cosine similarity is the most commonly used similarity metric in CLIP and relevant literature [3,4].
>
> We take the maximum similarity value across the text and visual features because if two classes have high text feature similarity but low visual similarity, misclassification can still occur, and vice versa. By selecting the maximum value, we ensure that our method accounts for class pairs where concept confusion is most likely to happen.
>
> [3] Learning Transferable Visual Models From Natural Language Supervision. In Proc. of ICML, 2021.
> [4] Does CLIP's Generalization Performance Mainly Stem from High Train-Test Similarity?. In Proc. of ICLR, 2024.
>
> ---
>
> We will incorporate all of the above experimental results and analyses in the next version.

---

> > ### Comment · Reviewer_d39N · 2025-04-08
> >
> > Thank you for the thorough rebuttals. I’ve read all the reviews, comments, and responses carefully. I will maintain my rating.

---

### Decision · Program_Chairs · 2025-05-01

**Decision:**

Accept (poster)

**Comment:**

This paper aims to investigate the underlying causes of the class imbalance issues when generating pseudolabels by VLMs. This paper identifies two primary contributing factors, including concept mismatch and concept confusion. To address the two issues, this paper proposes a novel framework incorporating concept alignment and confusion-aware calibrated margin mechanisms, which would mitigate class imbalance by enhancing underperforming classes and promoting balanced predictions across categories. Experimental results demonstrate that the proposed method effectively enhances the accuracy and balance of pseudolabels, outperforming SOTA methods.

This paper finally receives 1 accept and 3 weak accept, which means all the reviews are positive. Addressing class imbalance when generating pseudolabels by VLMs is a crucial challenge, and this paper identifies two key factors and proposes an effective method. The contribution of this paper meets the ICML standard.

I commend the authors adopt the reviewers' constructive suggestions and add the provided new experimental results into the final version.